# The *LCORL* Locus Is under Selection in Large-Sized Pakistani Goat Breeds

**DOI:** 10.3390/genes11020168

**Published:** 2020-02-05

**Authors:** Rashid Saif, Jan Henkel, Vidhya Jagannathan, Cord Drögemüller, Christine Flury, Tosso Leeb

**Affiliations:** 1Institute of Genetics, Vetsuisse Faculty, University of Bern, 3001 Bern, Switzerland; rashid.saif37@gmail.com (R.S.); jan.henkel@vetsuisse.unibe.ch (J.H.); vidhya.jagannathan@vetsuisse.unibe.ch (V.J.); cord.droegemueller@vetsuisse.unibe.ch (C.D.); 2Institute of Biotechnology, Gulab Devi Educational Complex, Lahore 54000, Pakistan; 3School of Agricultural, Forest and Food Sciences, Bern University of Applied Sciences, 3052 Zollikofen, Switzerland; christine.flury@bfh.ch

**Keywords:** *Capra hircus*, whole-genome sequence, body size, height, stature, QTL, selection signature, animal breeding, meat production

## Abstract

Goat domestication and human selection for valued traits have formed diverse breeds with characteristic phenotypes. This process led to the fixation of causative genetic variants controlling breed-specific traits within regions of reduced genetic diversity—so-called “selection signatures”. We previously reported an analysis of selection signatures based on pooled whole-genome sequencing data of 20 goat breeds and bezoar goats. In the present study, we reanalyzed the data and focused on a subset of eight Pakistani goat breeds (Angora, Barbari, Beetal, Dera Din Panah, Kamori, Nachi, Pahari, Teddy). We identified 749 selection signatures based on reduced heterozygosity in these breeds. A search for signatures that are shared across large-sized goat breeds revealed that five medium-to-large-sized Pakistani goat breeds had a common selection signature on chromosome 6 in a region harboring the *LCORL* gene, which has been shown to modulate height or body size in several mammalian species. Fine-mapping of the region confirmed that all five goat breeds with the selection signature were nearly fixed for the same haplotype in a ~191 kb region spanning positions 37,747,447–37,938,449. From the pool sequencing data, we identified a frame-shifting single base insertion into an isoform-specific exon of *LCORL* as a potential candidate causal variant mediating the size-increasing effect. If this preliminary result can be confirmed in independent replication studies, genotyping of this variant might be used to improve breeding programs and the selection for stature in goats.

## 1. Introduction

There are 25 goat breeds in Pakistan [1] with an estimated population of 76.1 million heads, which are mostly raised for milk, meat, hair, and hide purposes. Goats produce ~940 thousand tons of milk annually in Pakistan and ~732 thousand tons of mutton (goat and sheep) [2]. Around 6.8 million people are involved in small ruminant farming in Pakistan and rearing goats under nomadic, transhumant, household and sedentary production systems [3]. Genomic selection and proper targeted breeding policies are considered promising tools for genetic improvement of this valued species. Body size is an important trait to investigate for the improvement of meat-purpose domestic animals. It is also marginally associated with milk production [4,5,6].

Human adult stature and skeletal frame size is controlled by more than 700 genes [7,8,9,10,11,12]. The *LCORL* gene encoding ligand-dependent nuclear receptor corepressor-like has been repeatedly found to be associated with human height [7,8,9,10,11,12]. LCORL (previously also called MLR1) is a putative transcription factor that utilizes a conserved helix-turn-helix motif for DNA binding [13]. There are several experimentally confirmed human *LCORL* transcript isoforms that encode vastly different proteins. Not all of these proteins share the DNA-binding domain, and they also differ in their other protein domains [14].

Size variation in livestock species is typically controlled by fewer genes with larger effects as compared to humans [15,16]. The *LCORL* locus has also been found associated with height and body size in many domestic animal species, including dogs [17], horses [18,19,20], pigs [21], and sheep [22,23].

In cattle, the genomic locus is also strongly associated with height and growth-related traits. However, the functional effect in cattle has been claimed to be mediated by the *NCAPG* gene, which is adjacent to *LCORL* [16,24,25,26,27]. Determining which of these two genes is responsible for variability in height has not been possible because of the close proximity of these genes and the high levels of linkage disequilibrium among markers in this genome region. Recently, the identification of a missense variant in bovine *LCOR*—a gene with very high homology and potentially similar function to *LCORL*—as being associated with stature provides some supporting evidence for *LCORL* as the causative gene [28].

We previously conducted a comprehensive screen for selection signatures in 20 genetically diverse goat breeds [29]. In the current study, we used this dataset to analyze loci under selection in large-sized Pakistani goat breeds with the aim of identifying important genes for meat production.

## 2. Materials and Methods 

### 2.1. Ethics Statement

All goats in this study were privately owned and samples were collected with the consent of their owners. Extracted DNA from Pakistani goats was sent to Switzerland for downstream sequencing and genotyping. 

### 2.2. Animal Selection

Eight Pakistani domestic goat breeds selected for this study belong to diverse geographical regions across Pakistan. Breed averages of the height at withers served as proxies for the size phenotypes (Table 1). Peripheral blood samples were collected from goats of the selected breeds in EDTA vacutainers and stored at −20 °C.

Based on the personal experience of the authors and the recorded average wither height in the FAO database, Beetal (BEE), Dera Din Panah (DDP), and Kamori (KAM) were arbitrarily classified as large-sized breeds for this study. Angora (ANG), Barbari (BAR), Nachi (NAC), and Pahari (PAH) goats were considered medium in size, while Teddy (TED) goats were classified as small. Representative animals of each breed are shown in Figure 1.

### 2.3. DNA Extraction and Identification of Selection Signatures

DNA was extracted from whole blood using TIANamp Genomic DNA Kit (Tiangen Biotech, Beijing, China) according to the manufacturer’s instructions. Pooled sequencing (pool-seq) and identification of selection signatures of these samples has been described in detail before [29].

Briefly, pooled heterozygosity scoring statistics were used as described [21,30] to identify the regions under selection, which was calculated as $H_{p}=\frac{2\sum nMaj\sum nMin}{(\sum nMaj+\sum nMin{)}^{2}}$, where ∑nMaj and ∑nMin are the sums of major and minor alleles for all single-nucleotide polymorphisms (SNPs) in each window. Then, individual $H_{p}$ values were Z-transformed by $ZH_{p}=\frac{\left(H_{p}-\mu H_{p}\right)}{\sigma H_{p}}$, where $\mu H_{p}$ is the overall average heterozygosity, and $\sigma H_{p}$ is the standard deviation of all windows within each pool. We calculated $ZH_{p}$ and converted it to −$ZH_{p}$ values in 150-kb sliding windows. In this study, we applied a threshold of $-ZH_{p}$ ≥ 4 for putative selection signatures. The underlying sequence data are publicly available under the study accession PRJEB23815 at the European Nucleotide Archive.

### 2.4. Fine Mapping

A selection window of 150 kb with $-ZH_{p}$ > 4 was found common in three large and two medium-sized breeds, but was not present in the small breed group. In order to define the precise boundaries of the selection signature, we considered all variants in a 375 kb region spanning positions 37,650,001–38,025,000 on chromosome 6. We calculated the frequency of the minor allele at each variant in each breed pool. The selection signature was then defined as the region in which the five breeds with the selection signature showed a long consecutive stretch of variants with minor allele frequency < 0.2.

### 2.5. Gene Analysis

We used the ARS1 goat reference genome assembly [31], accession number: GCF_001704415.1, as reference for all downstream analyses. Numbering within the goat *LCORL* gene corresponds to the NCBI RefSeq accession numbers XM_018049322.1 (mRNA) and XP_017904811.1 (protein).

### 2.6. Sanger Sequencing

Seven *LCORL* variants were genotyped by direct Sanger sequencing after PCR amplification with the primers listed in Table 2.

The resulting amplicons were sequenced on an ABI 3730 DNA Analyzer after treating with exonuclease I and alkaline phosphatase. Finally, the obtained Sanger sequences were analyzed using the Sequencher 5.1 software (GeneCodes, Ann Arbor, MI, USA).

### 2.7. Association Analysis

We performed an allelic association study in 74 goats (Appendix A) using the genotypes at the seven genotyped variants in a case–control design and the PLINK 1.07 software [32]. We considered goats from BAR (*n* = 8), BEE (*n* = 12), DDP (*n* = 12), KAM (*n* = 13), and NAC (*n* = 12) as cases and goats from ANG (*n* = 6), PAH (*n* = 5), and TED (*n* = 6) as controls (Appendix A).

## 3. Results

### 3.1. Selection Signatures in Large-Sized Goats

Our pool-seq dataset comprised eight Pakistani goat breeds [28]. Each breed pool was composed of 12 animals except Angora, for which only 10 animals were contained in the pool. Pools were sequenced to 30× coverage and sequence data mapped to the ARS1 goat reference genome. On average ~12.7 million single nucleotide variants (SNVs) were observed in each breed from Pakistan as compared to ~11.7 million SNVs in Swiss goat breeds [29].

In the Pakistani goat breeds we identified a total of 2064 windows with $-ZH_{p}$ > 4. After merging overlapping windows, this resulted in 749 putative selection signatures in the eight studied breeds (Table 3, Appendix A).

In order to identify selection signatures related to body size, we searched for signatures that were shared between all three large-sized goat breeds in the study (BEE, DDP, KAM). This identified 18 common selection signatures in these breeds (Figure 2; Appendix A).

One of these shared selection signatures was located on chromosome 6 at ~38 Mb and contained the *LCORL* gene, which has been shown to be associated with size in many species. Interestingly, two of the medium-sized breeds, BAR and NAC, also had this selection signature harboring the *LCORL* gene. Inspection of the pool-seq data revealed that all five breeds shared the same major haplotype in this region. This prompted us to hypothesize that a size-increasing allele caused by the same genetic variant and identical by descent (IBD) is under selection in the BAR, BEE, DDP, KAM, and NAC goat breeds.

### 3.2. Fine-Mapping of the LCORL Selection Signature 

The window with the highest $-ZH_{p}$ score spanned positions 37.80–37.95 Mb in all five breed-pools. In order to precisely define the boundaries of the selection signature, we also considered adjacent regions with slightly elevated $-ZH_{p}$ scores and initially examined a 375-kb region spanning 37.600–38.025 Mb. Our dataset with all eight breed pools contained 1469 SNVs in this interval. We counted the minor and major alleles for each of these variants in each breed pool (Appendix A). A consecutive stretch of 156 variants showed reduced variation with minor allele frequencies of less than 0.2 in the five breeds with the selection signature. The five breeds under selection were nearly fixed for a shared 191-kb haplotype ranging from 37,747,447 to 37,938,449 (Figure 3A; Appendix A). We considered this as the critical interval harboring the hypothetical size-increasing genetic variant.

### 3.3. Search for Candidate Causative Variants

The critical interval at this selection signature contained four protein-coding genes (*FAM184B, DCAF16, NCAPG, LCORL*) and one gene for a non-coding RNA (LOC106502187; Figure 3B). Visual inspection of the short read alignments did not reveal any structural variants that were private to the five breed pools with the selected haplotype. Based on literature data in other mammalian species [16,17,18,19,20,21,22,23,24], we considered *LCORL* the most likely functional candidate gene for size. We selected six SNVs and a single base insertion that were located in potential exons of the *LCORL* gene and genotyped them on 74 individual goats. We calculated the association to the phenotype and observed the most significant association for *LCORL*:c.828_829insA (Table 5). The insertion allele had a frequency of 0.95 in the five breeds with the selection signature vs. 0.35 in three goat breeds that are not specifically selected for size. This variant introduces a frameshift in three of the five annotated transcripts. On the longest protein isoforms, the single-base insertion, is predicted to result in the truncation of more than 85% of the open reading frame. The formal variant designation on the protein level is XP_017904811.1:p.(Ser277Ilefs*38).

## 4. Discussion

We investigated selection signatures potentially related to body size due to the prime importance of this trait in the highly meat consuming society of Pakistan, which is influenced by religious and other personal choices as well as economic needs for marginal farmers in low-income countries [2,33,34,35]. Several Pakistani goat breeds have already been genotyped under the AdaptMap project for exploring goat diversity (e.g., local adaption and coat color genetics) [36,37,38], but to our knowledge the trait of skeletal frame size and adult height has not been addressed before.

Our comprehensive study identified 749 selection signatures in Pakistani goat breeds. We observed many selection signatures harboring genes that influence height or body size in other species, thereby validating our experimental approach. Examples include the *WARS2* gene, which is under selection in BAR, BEE, DDP, KAM, and NAC and associated with body fat distribution in humans [39], or *ADAMTS6*, which is under selection in BEE, DDP, KAM, and NAC and associated with bone length in mice [40].

The *LCORL* locus has previously been found to be associated with height in at least six mammalian species [16,17,18,19,20,21,22,23,24,25,26,27]. It is therefore not surprising that this locus is apparently also under selection in several medium-to-large-sized Pakistani goat breeds. However, it should be noted that this region does not show any evidence of selection in any of the previously studied Swiss goat breeds [29]. While there is accumulating evidence that genetic variation at the *LCORL* locus is involved in the determination of height and body size in diverse mammalian species, the molecular mechanism remains elusive. The strongest associations are frequently seen within a region harboring *DCAF16*, *NCAPG*, and the 3′–end of the *LCORL* gene. The in vivo function of the *LCORL* gene is not fully understood. *LCORL* encodes several different transcript isoforms, which mostly differ by the alternative use of at least four different exons at the 3′-end of the gene. One of them is almost 5 kb in length and predicted to encode more than 1600 amino acids. A single-base insertion, XM_022416410.1:c.3661_3662_insA, in this giant exon has been postulated to increase size in dogs [17].

In our study, another frame-shifting single-base-insertion variant in this exon showed the strongest differentiation between large and small breeds. It is intriguing that two independent single-base insertions into homologous exons in dogs and goats are associated with large body size. Unfortunately, there is no available functional confirmation for the causality of the insertion in dogs [17], and we did not perform any attempts to analyze the functional mechanism by which this *LCORL* variant might lead to an increase in body size. Nonetheless, in light of the parallels between dogs and goats, it is tempting to speculate that *LCORL*:c.828_829insA might be the true causative variant for a size-increasing QTL in Pakistani goats. Unfortunately, our study relying on breed-average phenotypes cannot provide conclusive proof for this hypothesis. Further studies are required to confirm this preliminary finding. As some medium-sized goat breeds are apparently not fixed at the *LCORL* locus, the next logical step will be a within-breed association study correlating individually measured height phenotypes with the genotypes at c.828_829insA in a breed that segregates both alleles. We must also caution that our analysis exclusively focused on the *LCORL* gene and that we cannot exclude an additional functional effect by any of the other three genes in the selection signature.

To the best of our knowledge, this is one of the first studies to report body-size-related selection signatures in different Pakistani goat breeds. The provided pooled heterozygosity statistics may also be used to investigate other breed-specific selection signatures and traits. This should help to enable more efficient breeding strategies.

## 5. Conclusions

A total of 749 selection signatures were observed in Pakistani goat breeds using pooled heterozygosity statistics. One selection signature harbored the *LCORL* gene and was observed in three large-sized and two medium-sized breeds. Detailed analysis of the *LCORL* selection signature suggested that the effect may be mediated by a frame-shifting single base insertion into the giant exon encoding parts of the long LCORL protein isoforms. However, this preliminary finding requires additional confirmation before the causality of this particular variant should be considered proven.

## Figures and Tables

**Figure 1 genes-11-00168-f001:**
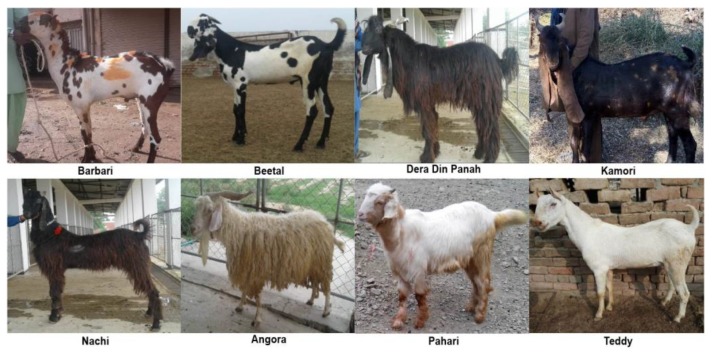
Representative animals of eight Pakistani goat breeds.

**Figure 2 genes-11-00168-f002:**
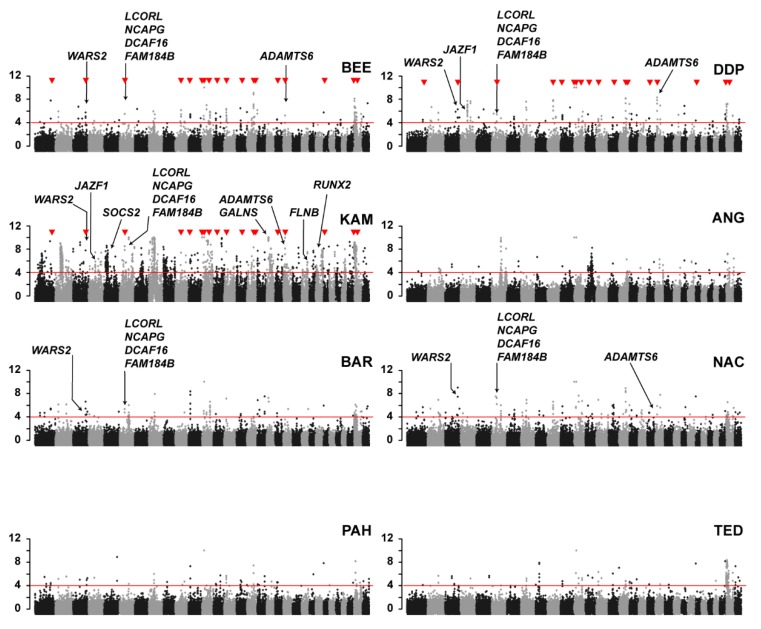
Manhattan plots illustrating the observed selection signatures in eight Pakistani goat breeds. Red triangles indicate 18 selection signatures that are shared between three breeds of large-sized goats (BEE, DDP, KAM). Selection signatures that harbor genes known to be related to height or body size in other species are indicated.

**Figure 3 genes-11-00168-f003:**
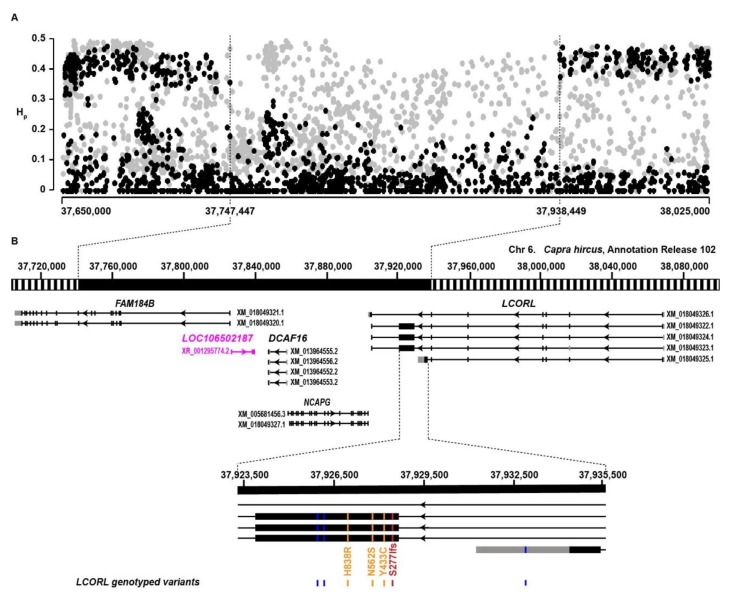
(**A**) Pooled heterozygosity (Hp) distribution at the *LCORL* locus on chromosome 6 in five medium/large-sized Pakistani goat breeds (black dots) and three small/medium breeds (grey dots). A ~191 kb region from 37,747,447 to 37,938,449 showed greatly reduced heterozygosity in the breeds with the selection signature. (**B**) The gene annotation for the selection signature is illustrated (NCBI annotation release 102). The genomic locations of seven genotyped variants in the *LCORL* gene are indicated in red (putative frameshift variant), orange (putative missense variants), or blue (silent or non-coding variants).

**Table 1 genes-11-00168-t001:** Phenotypic characteristics of eight goat breeds of Pakistan.

Breed Name	Abbr.	WH (cm)	BW (kg)	Main Purposes	Geographic Distribution	Population Size (2006) ^a^
Angora	ANG	75	47	Meat, hair	Punjab	NA
Barbari (Bari)	BAR	68	30	Meat, milk	Punjab, Sindh	2306
Beetal	BEE	NA	NA	Meat, milk	Punjab	4214
Dera Din Panah	DDP	81	61.7	Meat, milk, hair	Punjab	1424
Kamori	KAM	90	55	Meat, milk	Sindh	5294
Nachi (Bikaneri)	NAC	71	35	Meat, milk, hair	Punjab	1135
Pahari (Kajli)	PAH	77	67	Meat, milk, hair	Punjab, Balochistan	NA
Teddy	TED	64	33.9	Meat, milk	Punjab, Azad Jammu and Kashmir	1342

WH = wither height, BW = body weight; data taken from FAO [http://www.fao.org/dad-is/data]. ^a^ Thousand heads.

**Table 2 genes-11-00168-t002:** Primer sequences to amplify seven *LCORL* variants. The first amplicon contains two variable positions.

Primer Name	Primer Sequence	Product (bp)	Amplified Variants
LCORL_1F	CTTTCACCCAAGTCAGTGTCA	332	c.3480C>T
LCORL_1R	CCCCAGGTTGTGAAACAGAT		c.3360G>A
LCORL_2F	TTGGATGCTTTATACCCTTCTGA	213	c.2513A>G
LCORL_2R	AAAATCCCCTAAGGC CAAAA		
LCORL_3F	CATGTTGACTCAGCAATTCCA	226	c.1685A>G
LCORL_3R	ACAAATCAT GAAAAGGGTGAAAC		
LCORL_4F	TGCTGGTGTCAGAGATGGAG	215	c.1298A>G
LCORL_4R	CAGGCTTTCAGAGTCCTCGT		
LCORL_5F	AACAGCAAAGAGAAGCAGCA	495	c.828_829insA
LCORL_5R	TCCTTCTGAAGCACTTTCCA		
LCORL_6F	GGGTTCAGTATAGATCTGAGAGACC	479	c.777-4235T>C
LCORL_6R	TGGGCAGTGCATTTTAACTTT		

**Table 3 genes-11-00168-t003:** Selection signatures in the Pakistani goat breeds.

Breed	Animals per Pool	Windows under Selection	Selection Signatures	Genes in Selection Signatures
ANG	10	140	61	144
BAR	12	95	51	109
BEE	12	126	58	142
DDP	12	132	66	178
KAM	12	1244	341	889
NAC	12	130	79	137
PAH	12	61	44	92
TED	12	136	49	120
Total		2064	749	1811

**Table 5 genes-11-00168-t005:** Genotype phenotype association of selected SNVs in the *LCORL* gene.

Chr.	Position	cDNA Variant XM_018049322.1	Protein Variant XP_017904811.1	Alternative Allele Frequency in Breeds with the Selection Signature ^a^	Alternative Allele Frequency in Breeds without the Selection Signature ^b^	*p*-Value
6	37,925,990	c.3480C>T	p.=	0.05	0.26	3.0 × 10^−04^
6	37,926,110	c.3360G>A	p.=	0.93	0.35	2.1 × 10^−13^
6	37,926,957	c.2513A>G	p.His838Arg	1.00	1.00	NA
6	37,927,785	c.1685A>G	p.Asn562Ser	0.01	0.26	1.8 × 10^−07^
6	37,928,172	c.1298A>G	p.Tyr433Cys	0.96	0.53	3.9 × 10^−10^
6	37,928,645	c.828_829insA	p.Ser277Ilefs*38	0.95	0.35	8.0 × 10^−15^
6	37,932,928	c.777-4235T>C	intronic/3′-UTR of short isoform	0.01	0.09	1.0 × 10^−02^

^a^ BAR, BEE, DDP, KAM; NAC, ^b^ ANG, PAH, TED.

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
