# Peer review of "The LCORL Locus Is under Selection in Large-Sized Pakistani Goat Breeds"

_genes, 2020, doi:10.3390/genes11020168_

Round 1

Reviewer 1 Report

English language and style - I am not a native English speaker but I have not spotted anything wrong.

Originality/Novelty - novel is the Pakistani population(s) and specific findings; locus (loci are many) and tools are standard.

Significance of Content - high for the genomic selection, designing of microchips.

Quality of Presentation - clearly understandable.

Scientific Soundness - new specific insight; does not explain universal phenomena.

Interest to the readers - for breeders: possible enhancement of selection outcome; for the researchers: verification of the present results as they are important but still a hint.

I also attach your text with my few suggestions placed as notes or highlights. May be you will find them useful.

Author Response

We thank the reviewer for the annotated comments in the manuscript pdf-file and adressed them as required.

Reviewer 2 Report

Please clearly explain in the main text how to select large-sized breeds (probably based on wither height), and also show the (average) age for the data in Table 1.

L193: "Is therefore ~~" → maybe "This/It is therefore ~~"

Author Response

(1)

Please clearly explain in the main text how to select large-sized breeds (probably based on wither height).

Response: We indeed selected large-sized breeds based on wither height. We tried to give an objective measure for the breed average wither height by giving the FAO data in table 1. We revised lines 74-76 in the main text to make this clear.

(2)

Show the (average) age for the data in Table 1.

Response: Unfortunately, the average age for maturity of Pakistani goat breeds is not available in the FAO database.

(3)

L193: "Is therefore ~~" → maybe "This/It is therefore ~~"

Response: Revised accordingly.